# Genome-Wide Bioinformatics Analysis of *SWEET* Gene Family and Expression Verification of Candidate *PaSWEET* Genes in *Potentilla anserina*

**DOI:** 10.3390/plants13030406

**Published:** 2024-01-30

**Authors:** Javed Iqbal, Wuhua Zhang, Yingdong Fan, Jie Dong, Yangyang Xie, Ronghui Li, Tao Yang, Jinzhu Zhang, Daidi Che

**Affiliations:** 1College of Horticulture and Landscape Architecture, Northeast Agricultural University, Harbin 150030, China; iqbal468@aup.edu.pk (J.I.); 15346549367@163.com (W.Z.); fanying_dong@163.com (Y.F.); xieyangyang0715@163.com (Y.X.); lronghui0927@163.com (R.L.); yangtao@neau.edu.cn (T.Y.); jinzhuzhang@neau.edu.cn (J.Z.); 2Key Laboratory of Cold Region Landscape Plants and Applications, Harbin 150030, China

**Keywords:** *Potentilla anserina*, *SWEET*, swollen, tuber, genome-wide identification

## Abstract

Sugars act as the main energy sources in many fruit and vegetable crops. The biosynthesis and transportation of sugars are crucial and especially contribute to growth and development. *SWEET* is an important gene family that plays a vital role in plants’ growth, development, and adaptation to various types of stresses (biotic and abiotic). Although *SWEET* genes have been identified in numerous plant species, there is no information on *SWEETs* in *Potentilla anserina*. In the present study, we performed a comprehensive genome-wide bioinformatics analysis and identified a total of 23 candidate *PaSWEETs* genes in the *Potentilla anserina* genome, which were randomly distributed on ten different chromosomes. The phylogenetic analysis, chromosomal location, gene structure, specific cis-elements, protein interaction network, and physiological characteristics of these genes were systematically examined. The identified results of the phylogenetic relationship with *Arabidopsis thaliana* revealed that these *PaSWEET* genes were divided into four clades (I, II, III, and IV). Moreover, tissue-specific gene expression through quantitative real-time polymerase chain reaction (qRT-PCR) validation exposed that the identified *PaSWEETs* were differentially expressed in various tissues (roots, stems, leaves, and flowers). Mainly, the relative fold gene expression in swollen and unswollen tubers effectively revealed that *PaSWEETs* (7, 9, and 12) were highly expressed (300-, 120-, and 100-fold) in swollen tubers. To further elucidate the function of *PaSWEETs* (7, 9, and 12), their subcellular location was confirmed by inserting them into tobacco leaves, and it was noted that these genes were present on the cell membrane. On the basis of the overall results, it is suggested that *PaSWEETs* (7, 9, and 12) are the candidate genes involved in swollen tuber formation in *P. anserina*. In crux, we speculated that our study provides a valuable theoretical base for further in-depth function analysis of the *PaSWEET* gene family and their role in tuber development and further enhancing the molecular breeding of *Potentilla anserina.*

## 1. Introduction

All plants depend on photosynthesis, due to which plants produce their food (e.g., sugars and starch) mainly in source tissue (e.g., leaves) and transport this food to the sink organs (e.g., flowers, fruits, and roots) via a long-distance pathway to sustain the growth and development of these sink organs [1]. Sucrose is considered to be the primary carbohydrate in most plants and is transported through symplastic and apoplastic pathways, in which sugar transporters play significant roles [2]. Until now, a total of three families of sugar transporters have been recognized, comprising monosaccharide transporters (MSTs), sucrose transporters (SUTs), and sugar that will eventually be exported via transporters (*SWEETs*) [3].

*SWEETs* are a novel type of sugar carrier that was first recognized in *Arabidopsis thaliana* (a model plant) in 2010, mostly facilitating sugar transport [4]. Due to the advanced genome sequencing technology in plants, *SWEETs* genes have been studied in various plants, such as *Arabidopsis thaliana* [4], *Glycine max* [5], *Gossypium hirsutum* [6], *Triticum aestium* [7], *Vitis vinifera* [8], *Sorghum bicolor* [9], *Ipomoea batatas* [10], and *Musa acuminata* [11]. The *SWEETs* gene family is divided into four clades, and each clade has a diverse preference for the transport of monosaccharides or disaccharides. In general, clades I and II prefer to transport hexose, while clade III prefers to transport mainly sucrose. In addition, clade IV tends to transport mainly fructose [12]. As membrane-bound proteins, *SWEETs* contain seven transmembrane domains (7TMs) in eukaryotes, and these 7TMs form two parallel three-helix bundles that are connected by one central transmembrane [13].

*SWEETs* play a vital role in the transport of sugar across plasma membranes and intracellular membranes in prokaryotes and eukaryotes [14]. SWEETs proteins are more proficient of transporting sugar in two-way directions without energy requirements than those by sucrose transporters (SUT) and monosaccharide transporters (MST) [15]. Most of the studies revealed that *SWEETs* genes are involved in various physiological processes in plants, comprising phloem development, nectar secretion [16], seed development, phloem loading, fruit development, and other biotic and abiotic stresses [17]. Sugar is an important factor that determines the fruit quality and yield, and the transport and accumulation of sugar are mainly regulated by sugar transporters [18]. The various experiments exposed that *SWEETs* are mainly involved in the sugar accumulation in fruits and vegetables. *ClSWEET3* helped in the uptake of hexose from the intercellular space to the fruit in *Citrullus lanatus*, and overexpression of *ClSWEET3* could enhance sugar content and thus improve their quality [19]. Due to the fact that *SWEET* genes are important in sugar allocation, several *SWEET* genes were subjected to artificial selection during crop domestication.

SWEET4 proteins in rice and maize, which facilitate hexose transport across endosperm, were strongly chosen throughout domestication to support the development of cereal grains [20]. In the *Prunus salicina*, it is noted that *SWEETs* genes might have potential to transport glucose and fructose during fruit development [21]. Sucrose is produced in the source organ and transported to the sink; it is mainly regulated by *SWEETs*, such as *CitSWEET11* and *PuSWEET15*, which are significantly involved in the transport of sucrose in citrus and pear fruit, respectively [22]. *SWEETs* regulate the sugar transport and thus act as an important factor in the formation of fruit, seed, and tubers. *AtSWEET11* and *AtSWEET12* are critical transporters for seed filling in Arabidopsis [23]. *ZmSWEET5* in *Zea mays* and *OsSWEET11* in *Oryza sativa* have a significant role in the transport of sucrose to the endosperm to stimulate seed filling. It was reported that GmSWEET15 in soybeans facilitated sucrose transport to embryos to support seed development [24]. In addition, *SWEETs* play an important role in regulating the effects of various types of biotic and abiotic stress. It is noted in Arabidopsis that *AtSWEET2* restrains sugar transport, probably by reducing the availability of glucose from the cytosol in to the vacuole, thus controlling carbon loss to the rhizosphere and contributing to enhanced resistance to *Pythium* [25]. *AtSWEET16* and *CsSWEET2* were noted to improve the freezing tolerance of transgenic plants in Arabidopsis and cucumber, respectively [26].

*Potentilla anserina*, which belongs to the family Rosaceae, is an herbaceous, perennial, stoloniferous plant with edible tuberous roots that is widely distributed in temperate zones around the world [27]. *P. anserina* has served as an important food and medicine source over thousands of years, and the tuberous roots of this plant have been applied in various herbal medicines due to the fact that they promote body fluid production, thus relieving thirst, strengthening the stomach and spleen, and invigorating the blood [28]. In ancient times, the whole plant of *P. anserina* was used as Chinese medicine for hematemesis treatment. The remedies are widely used in various folk and medical systems, particularly in traditional Tibetan medicine and folk medicine [29]. Moreover, they are rich in polysaccharides and saponins, which can resist viruses and enhance immunity. In addition, they also contain various amounts of trace elements (calcium, potassium, zinc, and magnesium), which are necessary for the human body [30]. The development and thickening of tuberous roots is one of the most important processes that determine the yield and quality of *P. anserina*. It has been stated in previous studies of tuberous/root crops (sweet potato and radish) that *SWEETs* plays an important role in the growth and development of tubers/roots [31,32].

In this study, we performed a genome-wide bioinformatics analysis of the *SWEETs* gene family and identification of major genes regulating tuber formation in *P. anserina*. We systematically explored the physicochemical properties of proteins, chromosomal localization, phylogenetic comparisons, cis-acting promoters, and their expression patterns between swollen and unswollen roots. The purpose of this study was to provide an understanding of the *PaSWEETs* gene family and their role in tuber formation. We believe that our research findings will provide a strong base for future research regarding the *SWEET* genes and their possible role in tuber formation in various tuberous/roots crops and will assist in genetic improvement of horticulture traits using advanced breeding programs.

## 2. Results

### 2.1. Characterization of P. anserina SWEETs Family Genes

A total of 23 *SWEET* gene family members were identified at different chromosomal locations on the *P. anserina* genome (PRJNA640225), and these genes were named from *PaSWEET1* to *PaSWEET23*. The physicochemical properties of the 23 *SWEET* genes in *P. anserina* indicated that the CDS length ranged from 693 bp (*PaSWEET23*) to 960 bp (*PaSWEET4*) (Table 1). The amino acid (aa) lengths of *PaSWEETs* varied from 230 aa (*PaSWEET23*) to 319 aa (*PaSWEET4*). The maximum molecular weight (35.47 KDa) was noted for *PaSWEET4*, while the lowest molecular weight (25.94 KDa) was noted for *SWEET12*. In the case of the isoelectric point (PI), 82% of the members had a PI greater than 7, which indicated that they were basic proteins, while 18% had a PI below 7, which indicated that they were acidic proteins. The highest PI was noted for *PaSWEET1* (9.62 PI), while the lowest PI was noted for *PaSWEET14* (5.41). The subcellular localization showed that all *PaSWEETs* genes were located in the cell membrane.

### 2.2. Analysis of Chromosomal Location of the PaSWEET Genes

The chromosomal location analysis indicated that a total of 23 *PaSWEET* genes were randomly distributed on ten chromosomes of the *P. anserina* genome (Figure 1). The maximum number of genes were identified at Chr3, Chr4, Chr6, Chr9, and Chr10. *PaSWEET1*, *PaSWEET2*, and *PaSWEET3* genes existed on Chr3; *PaSWEET4*, *PaSWEET5*, and *PaSWEET6* were located on Chr4; *PaSWEET9*, *PaSWEET10*, and *PaSWEET11* were positioned on Chr6; and *PaSWEET15*, *PaSWEET16*, *PaSWEET17*, *PaSWEET18*, *PaSWEET19*, and *PaSWEET20* were located on Chr9 and Chr10. The minimum number of genes (one) was found on Chr8 (*PaSWEET14*) and one gene (*PaSWEET21*) on Chr12, but there were no genes present on Chr 1, 2, 11, and 14. Moreover, there were two genes each on Chr5 and Chr13, which were *PaSWEET17* and *PaSWEET18*, and *PaSWEET22* and *PaSWEET23*, respectively.

### 2.3. Phylogenetic Analysis of Conserved Motif and Gene Structure of SWEET Genes

For a better understanding of the evolution of *PaSWEET* genes, a phylogenetic tree was constructed by combining 23 *PaSWEET* genes with 17 *SWEET* genes of *Arabidopsis thaliana* (*AtSWEETs*) that were previously reported. The analysis of these 40 *SWEET* genes (23 *PaSWEET* + 17 *AtSWEETs*) through a phylogenetic tree indicated that they were divided into four main clades (Clade I, Clade II, Clade III, and Clade IV) (Figure 2). The detailed distribution of these *SWEETs* in each clade is as follows: Clade I (9, 3), Clade II (2, 2), Clade III (4, 5), and Clade IV (8, 7). A total of 23 identified *PaSWEET* genes were distributed into four different clades.

In addition, the specific numbers of *PaSWEET* genes in each clade (I, II, III, and IV) were 9, 2, 4, and 8, respectively (Figure 3A). In order to further identify the PaSWEET protein sequence, we predicted their conserved domains (Figure 3B). After analyzing the protein sequences of 23 *PaSWEETs*, ten consensus motifs were identified, and only two proteins (*PaSWEET16* and *PaSWEET19*) containing nine motifs were observed (Figure 3B). Moreover, 43.47% of *PaSWEETs* contained six motifs, 21.73% of *PaSWEETs* contained five motifs, and 21.73% of *PaSWEETs* contained seven motifs. In addition, there was only one gene (*PaSWEET21*)-related protein, which contained four motifs. It was noticed that *PaSWEETs* in the same clade except for some proteins were similar in number and location of motifs, which suggested that they may have similar encoding functions in regulations of plant growth and development.

To better understand the gene structure of *PaSWEETs*, we analyzed their exon and intron structures (Figure 3C). It can be observed that most of the *PaSWEET* genes (*PaSWEET1*, *PaSWEET2*, *PaSWEET3*, *PaSWEET4*, *PaSWEET5*, *PaSWEET6*, *PaSWEET7*, *PaSWEET8*, *PaSWEET9*, *PaSWEET10*, *PaSWEET11*, *PaSWEET12*, *PaSWEET13*, *PaSWEET14*, *PaSWEET15*, *PaSWEET16*, *PaSWEET18*, *PaSWEET19*, and *PaSWEET22*) contained six exons (ranging from 0 to 6). Moreover, three *PaSWEETs* (*PaSWEET17*, *PaSWEET20*, and *PaSWEET21*) contained five exons in their structure, whereas only one gene (*PaSWEET23*) had four exons. These findings suggested that most of the *PaSWEETs* genes had similar gene structures, which further indicated that *SWEETs* were evolutionarily conserved in the *P. anserina* genome.

### 2.4. Analysis of Cis-Regulatory Element in the Promoter Region of PaSWEETs

In order to explore the in-depth regulatory mechanism by which *PaSWEET* genes impact plant growth and development, and biotic and abiotic stresses, we performed a cis-element analysis by submitting the upstream sequence (2000 bp) of the translation site to the PlantCare database to check for the existence of particular cis-elements. It was noted that the promoter region of *PaSWEETs* consisted of 21 regulatory elements that played a vital role in plant growth and development and overcoming the biotic and abiotic stresses (Figure 4). Furthermore, it was noted that almost all of the *PaSWEET* gene contained cis-regulatory elements that were associated with plant hormones (abscisic acid, gibberellin, and methyl jasmonate), and these hormones played a specific role in the development of tubers and stimulating cell division and expansion. It also indicated that light-responsiveness elements were rich in the promoter regions of *PaSWEETs* (Figure 3). These results further suggested that *PaSWEETs* played a key role by regulating plant growth and development, hormonal crosstalk, and biotic and abiotic stress adaptation in *P. anserina*.

### 2.5. Analysis of Relative Gene Expression of PaSWEETs Genes in Various Tissues

To examine the roles of *PaSWEET* genes in *P. anserina*, their relative gene expression was analyzed in various sampled tissues (roots, stems, flowers, and leaves) (Figure 5). The qRT-PCR analysis indicated that a total of 13 *PaSWEET* genes were found to be differentially expressed in all tissues. The result indicated that the maximum amount of *PaSWEETs* (*PaSWEET1*, *PaSWEET2*, *PaSWEET4*, *PaSWEET5*, *PaSWEET6*, *PaSWEET7*, *PaSWEET8*, *PaSWEET15*, *PaSWEET16*, *PaSWEET18*, *PaSWEET19*, *PaSWEE21*, and *PaSWEET22*) were highly expressed in flowers, which indicated that flowers might be the target tissue for future research on the function of *PaSWEETs*. The minimum amount of *PaSWEETs* (*PaSWEET9*, *PaSWEET11*, and *PaSWEET12*) was highly expressed in stems. In addition, *PaSWEET13*, *PaSWEET14*, and *PaSWEET20* genes were highly expressed in the root tissues. Moreover, *PaSWEET3*, *PaSWEET9*, *PaSWEET10*, *PaSWEET17*, and *PaSWEET23* were noted to be highly expressed in leaves. Thus, our expression analysis suggested that *PaSWEETs* had different roles to play in various tissues. 

### 2.6. Relative Expression of PaSWEETs Genes in Swollen and Unswollen Tubers

In order to explore the possible biological roles of these proteins in tuber growth and development, the qRT-PCR was used to assess the expression level of major *PaSWEETs* genes in both swollen and unswollen tubers of *P. anserina* (Figure 6). Gene expression analysis showed that swollen and unswollen tubers had different expression levels. Among all of the *PaSWEETs* genes, three *PaSWEETs* (*PaSWEET7*, *PaSWEET9*, and *PaSWEET12*) were found to be highly expressed in swollen tubers by 300-, 120-, and 100-fold more than in unswollen tubers, respectively. *PaSWEET15*, *PaSWEET18*, and *PaSWEET21* were also expressed in swollen tubers, but their expression levels were only 30-, 40-, and 15-fold, respectively. Moreover, there were many genes, i.e., *PaSWEET5*, *PaSWEET6*, *PaSWEET13*, *PaSWEET17*, *PaSWEET20*, and *PaSWEET22*, that were expressed in unswollen tubers, but their expression levels were very low. Based on these results, it is suggested that *PaSWEET7*, *PaSWEET9*, and *PaSWEET12* are involved in tuber growth and development.

### 2.7. Subcellular Location Analysis of PaSWEET Gene

By using the Cell-PLoc software package (version 2.0), (http://www.csbio.sjtu.edu.cn/bioinf/Cell-PLoc, accessed on 22 November 2023), it was predicated that all 23 *PaSWEETs* genes were present on the cell membrane. To further confirm the subcellular location, *PaSWEETs* genes (7, 9, and 12) were cloned and introduced to tobacco leaf for the identification of the subcellular location. The identified results indicated that 35S::*PaSWEET7*-GFP, 35S::*PaSWEET9*-GFP, and 35S::*PaSWEET12*-GFP were found to be present on the cell membrane (Figure 7), and hence, our results are consistent with the expected results as given in Table 1. Based on this result, it is suggested that *PaSWEETs* genes (7, 9, and 12) are membrane-localized and play a vital role in the transport of sugars into or out of the cells.

### 2.8. Protein Interaction Network of PaSWEETs in Potentilla anserina

In order to further explore the regulatory network of PaSWEETs, the PaSWEETs protein interaction network was constructed based on *Arabidopsis thaliana* proteins. The identified results revealed that proteins of *PaSWEET* genes (7, 9, and 12) might interact with various types of proteins, i.e., SUC2, SUC4, MSSP1, MSSP2, RPTIB, STP13, AVT3A, ESL1, and ARPC2B, to regulate plant growth and development, as well as play a vital role in biotic and abiotic stress adaptation in *P. anserina* (Figure 8).

## 3. Discussion

The *SWEET* gene plays a major role in the allocation of photoassimilation from the source organ (leaves) to the sink organ (fruits/tubers) and thus plays a tremendous role in the growth and development of these sink organs [33]. The *SWEET* gene family members have been identified in numerous crops such as apples [34], potatoes [35], tomatoes [36], and cabbage [37]; however, *P. anserina* has received little attention. The genome-wide identification of candidate *SWEETs* genes in *P. anserina* will provide a base for further research regarding the *SWEETs* genes and their possible role in tuber root formation.

In this study, 23 *PaSWEETs* genes were identified (Figure 1 and Table 1), which were categorized into four subfamilies (Clades I–IV), including nine members in Clade I, two in Clade II, four in Clade III, and eight in Clade IV (Figure 3A). Furthermore, the amount and kind of *SWEETs* distributed in each *P. anserina* subgroup differed from those found in Arabidopsis and many other plants (Figure 2). These findings suggest that the *SWEETs* gene in the terrestrial plant genome may have experienced lineage-specific diversity [38]. The gene structure analysis indicated that most of the *PaSWEETs* genes had six exons (Figure 3). These results are in line with those found in apples [39], litchi [40], and cucumbers [41]. In general, the addition or deletion of introns, as well as the pattern of exon–intron distribution, can alter the complexity of the gene structure, thereby resulting in novel gene functions, which are thought to be vital factors that affect gene family evolutionary mechanisms [42].

The analysis of cis-regulatory elements showed that a total of 21 cis-regulatory elements played a dynamic role in plant growth and development and in overcoming biotic and abiotic stress. Based on the genome-wide analysis, it was noted that almost all *PaSWEETs* contained cis-regulatory elements that were associated with plant hormones (abscisic acid, gibberellin, and methyl jasmonate). These results reflected that the *PaSWEET* gene family members may have a crosstalk with plant hormones to enhance tuber growth and development by facilitating cell division, as well as play a significant role in the plant’s tolerance against biotic and abiotic stress. The involvement of endogenous hormones is significant in the mechanism of tuberous root enlargement. It is noted that plant hormones such as abscisic acid and cytokinin play a significant role in the formation of tubers in various tuberous crops [43]. It is widely believed that zeatin plays a significant role in the initiation of tuberous roots through the activation of the primary cambium. The hormone abscisic acid governs the process of tuberous root thickening by stimulating the cell division of meristematic cells [44]. While studying the effect of plant hormones on the growth of tuberous roots, it was found that at the early stage of tuberous growth, auxin levels gradually increased, while they tended to decrease at the later stage. Furthermore, it is also noted that cytokinin and abscisic acid gradually increased throughout the tuberous growth period [45]. Moreover, it is also noted that most CREs in the *PaSWEETs* promoter were related to hormone and light response, and hence, it was revealed that the *SWEETs* gene plays a tremendous role in the response to abiotic stress. It has been reported that *SWEETs* play a role in plant responses to abiotic stresses including cold, salinity, and drought [46].

The relative gene expression of the *PaSWEET* gene in different sampled tissues (roots, stems, flowers, and leaves) showed different expression patterns. This might reflect that *PaSWEETs* have different roles in various tissues, e.g., the *AtSWEET12* genes are vital transporters in the Arabidopsis family, which are found in a subset of leaves and play a significant role in the phloem unloading process [47]. Moreover, *ZjSWEET2* in jujube [48] and *StSWEET11* in potato [49] were also proven to have a specific function in sugar loading. In this study, a total of 13 *PaSWEETs* genes were highly expressed in flowers, which is consistent with other previous studies. It has been reported that during the anthesis of floral organ in *Jasminum sambac* maximum (7) *SWEET* genes are expressed [50]. The *SWEET* gene family also plays key roles in the growth of flowers and fruits by facilitating the unloading of sugar in phloem. In addition, overexpression of *AtSWEET10* in Arabidopsis led to quick flowering, which might suggest the significance of sugar transport in the floral phase [51]. In addition, *PaSWEET10* and *PaSWEET17* were highly expressed in leaves, which is in line with the results [50]. *SWEET17* is known to play a significant role during the senescence of leaves and was noted to be up-regulated to remobilize carbohydrates during the senescence process [52]. Similarly, while working on day lily plant, *HfSWEET17* was found to be highly expressed in leaves, which further suggests their possible roles in leaf growth [53].

In this current study, it was found that *PaSWEETs* showed different expression patterns in swollen and unswollen tubers, and different *PaSWEETs* (*PaSWEET7*, *PaSWEET9*, and *PaSWEET12*) were found to be highly expressed in swollen tubers. So, it is suggested that *PaSWEETs* may be involved in the formation of tuberous roots by regulating the accumulation in *P. anserina*. Our results are consistent with those found in sweet potatoes, Arabidopsis, and radishes [31,32,54] and similarly revealed that *SWEETs* play different roles in tuberous root development, hormone crosstalk, and carotenoid accumulation in sweet potatoes and its relatives [31]. It has been reported that *SWEETs* contribute to assimilate accumulation and root development. *AtSWEET11* and *AtSWEET15* in Rice are key transporters during the seed filling stage [32]. It has been reported that *RsSWEET2b* and *RsSWEET17* in radishes are highly enriched in the roots, indicating their function in root growth and development, and overexpression of *RsSWEET17* in Arabidopsis showed longer roots, a higher amount of soluble sugar and maximum fresh weight. Moreover, *RsSWEETs* may have a role in the thickening of radish taproots by enhancing cambium activity mediated by soluble sugars [54]. *SWEETs*, which act as major transporters and play a tremendous role in the accumulation of sugar in sink organs, thus improve tuber/root development in *Ipomoea batatas* and *Raphanus sativus* [31,54].

Further, the subcellular localization indicated that *PaSWEETs* (7, 9, and 12) were found to be present on the cell membrane (Figure 7). Most of the studies showed that *SWEET* genes could be homo-oligomerized and hetero-oligomerized to create functional pores and provide channels for the transport of sugars. *SlSWEET7a* in tomatoes is localized on the plasma membrane, forms both homodimers and heterodimers, and generates pores for the transport of sugars, mainly for larger substrates such as fructose and sucrose [55].

The protein interaction network showed that PaSWEETs proteins interact with multiple proteins to regulate plant growth and development and plant defense responses (Figure 8). For example, *PaSWEETs* gene (7 and 9) proteins are related to SUC2 and SUC4, which are sucrose transporters in *Arabidopsis thaliana*, representing their vital roles in sugar transport to the sink organ [56]. Similarly, PaSWEETs interact with MSSP2, the monosaccharide transporter, which engages in the transport of monosaccharide from the source to the sink organ and has a significant role in the development of these sink organs [57]. The proteins of *PaSWEETs*, which are also related to STP13, would engage in the active resorption of hexoses in order to provide the additional energy required to initiate plant defense responses against abiotic stress [58]. Furthermore, the PaSWEET9 protein is related to ARPC, which has a significant role in enhancing the quantitative resistance mechanism in certain *A. thaliana* accessions toward diseases [59], and ESL, which is involved in responding to various types of abiotic stress (i.e., water stress) [60].

## 4. Materials and Methods

### 4.1. Genome-Wide Identification of SWEET Genes in Potentilla anserina

The NCBI genome database was searched for the genome sequence and annotation of *P. anserina*. The domain for the *SWEET* protein (PF03083) was obtained from the Pfam database and employed in conjunction with HMMER (3.3.2) software, in which the e-value was set as (e-value < 10^−5^) to identify the SWEET proteins of *P. anserina* [61]. By verifying the SWEET domain of *PaSWEETs* by ExPASy and Pfam [62] and removing any duplicate entries, each *PaSWEET* was given a unique name based on its position on the reference chromosomes. The PaSWEETs protein and genome sequences for *A. thaliana* were obtained from the Ensemble database [63]. The ProtParam tool (http://web.expasy.org/prot.param) was employed to assess the sequence length, isoelectric point (PI), and molecular weight (MW) of each PaSWEET protein. The Cell-PLoc was used for the identification of the subcellular localization of candidate *PaSWEETs* [64].

### 4.2. Chromosomal Location and Tandem Duplication Analysis

The information regarding the locations for *PaSWEET* genes was obtained from the GFF genome annotation of *P. anserina*, and the chromosomal visualization of *PaSWEET* genes was generated by using the Tbtools software, (version TBtools-II) [65].

### 4.3. Gene Structure, Conserved Motif, and Phylogenetic Association Analysis

The gene structures of *PaSWEET* genes were predicted based on the genomic and coding sequence by using the Gene Structure Display Server. The conserved motifs of full-length of PaSWEET proteins were identified by using MEME (https://meme-suite.org/meme/tools/meme, accessed on 22 November 2023). The phylogenetic tree of the *P. anserina* and *A. thaliana SWEETs* genes was created by MEGA X, (version 7.0) software based on the neighbor-joining method and 1000 bootstrap replications [66].

### 4.4. Cis-Acting Element Analysis

The 2000 bp upstream sequence of the coding region of *PaSWEETs* was submitted to the PlantCARE software, version 1.58 (https://bioinformatics.psb.ugent.be/webtools/plantcare/html, accessed on 22 November 2023), and cis-acting elements were identified [67]. 

### 4.5. Protein Interaction Analysis of PaSWEETs

Protein interaction networks of PaSWEETs were predicated by STRING (https://cn.string-db.org, accessed on 22 November 2023) based on *A. thaliana* protein, and the cytoscape (version 3.9.1) software was used to construct the network map [68].

### 4.6. Relative Gene Expression Analysis of PaSWEETs

The plants of *P. anserina* were grown in the plastic greenhouse at Xiangyang Agricultural Station of Northeast Agriculture University (NEAU), and roots, stems, leaves, flowers, and swollen and unswollen tubers were sampled, quickly put in liquid nitrogen, and snap-frozen at −80 °C for further experimentation. RNA from each sample tissue was extracted using the Total RNA Isolation Kit (Vazyme, Nanjing, China), and complementary DNA (cDNA) was synthesized from isolated RNA by using the cDNA synthesis kit (gDNA digester plus, Vazyme, Nanjing, China). The primer for *PaSWEET* genes was generated using the Primer Premier (version 5) Software [69]. For qRT-PCR reactions, we used BIO-RAD CFX96 (Bio-Rad, Hercules, CA, USA) with the Real Universal Taq Pro premix (SYBR Green) (Vazyme, Nanjing, China). A total of three biological replications of each sample were used for the subsequent analysis through qRT-PCR. The relative gene expression of *PaSWEETs* was checked by using the comparative 2^−ΔΔCT^ method. The information of all exported primer sequences of identified genes can be seen in Appendix A.

### 4.7. Subcellular Location Analysis of PaSWEET

The open reading frames (ORFs) of *PaSWEET7*, *PaSWEET9*, and *PaSWEET12* were cloned by using forward and reverse primers and then ligated to a vector (pCAMBIA1300-GFP). The untargeted GFP as an empty vector was used as a control. The recombinant plasmid and the empty vector were introduced to *Agrobacterium tumefaciens* GV3101 and transiently transfected into tobacco leaves with a syringe. After two days post infiltration, samples were taken from the tobacco leaves, the subcellular localizations of the PaSWEETs and positive control were visualized using confocal laser scanning microscopy (TCS SP8, Wetzlar, Germany) using a filter block to select for spectral emission at 488 nm, and images were captured.

### 4.8. Data Analysis

The numerical values of all the measured data were recorded on a regular basis. The significant differences among the obtained results were observed by performing the Duncan’s test at levels of *p* < 0.05 (significant) and *p* < 0.01 (highly significant) in the SPSS22 software (IBM SPSS Statistics ver. 19.0; IBM Corp., Armonk, NY, USA). The software GraphPad Prism (http://www.graphpad.com, version 9.0) was utilized for graphs visualization.

## 5. Conclusions

The present study effectively identified and analyzed a total of 23 *PaSWEET* genes (*PaSWEET1–PaSWEET23*) in the *Potentilla anserina* genome. To better understand the general features of the major *PaSWEETs* gene structure, we analyzed the exon–intron and conserved motif. Furthermore, we conducted a cis-element study in the promoter region of *PaSWEETs* for investigating the putative regulatory mechanism through which *PaSWEET* genes influenced plant growth and development. The tissue-specific gene expression analysis showed that all the *PaSWEETs* were differently expressed in different tissues (roots, stems, leaves, and flowers). In addition, we performed qRT-PCR to analyze the expression level of *PaSWEETs* in swollen and unswollen tubers, which showed that most of the *PaSWEET* genes were expressed in swollen tubers, but *PaSWEET* genes (7, 9, and 12) were highly expressed, suggesting the specific role in tuber growth and development. Finally, these *PaSWEETs* genes (7, 9, and 12) were inserted into tobacco leaves, which revealed the localization on the cell membrane.

## Figures and Tables

**Figure 1 plants-13-00406-f001:**
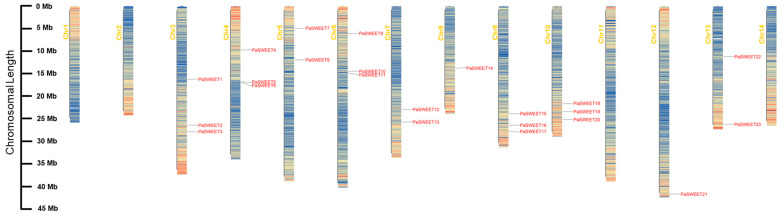
Chromosomal localization of *PaSWEETs* genes and their distribution on whole-genome chromosomes. Each bar represents chromosomes. The chromosome numbers are shown on the left side, and the gene names are presented on the right side. The location of each gene is displayed on the line bar.

**Figure 2 plants-13-00406-f002:**
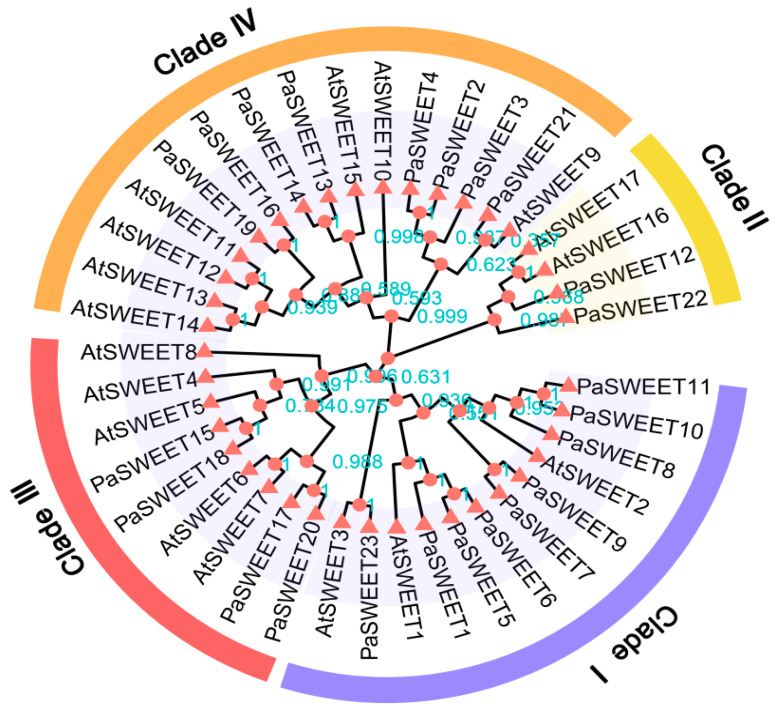
Phylogenetic tree of *SWEET* gene family members of *P. anserina* and *A. thaliana*, based on the neighbor joining (NJ) method and 1000 bootstrap replications. The four different clades are distinguished by different colors.

**Figure 3 plants-13-00406-f003:**
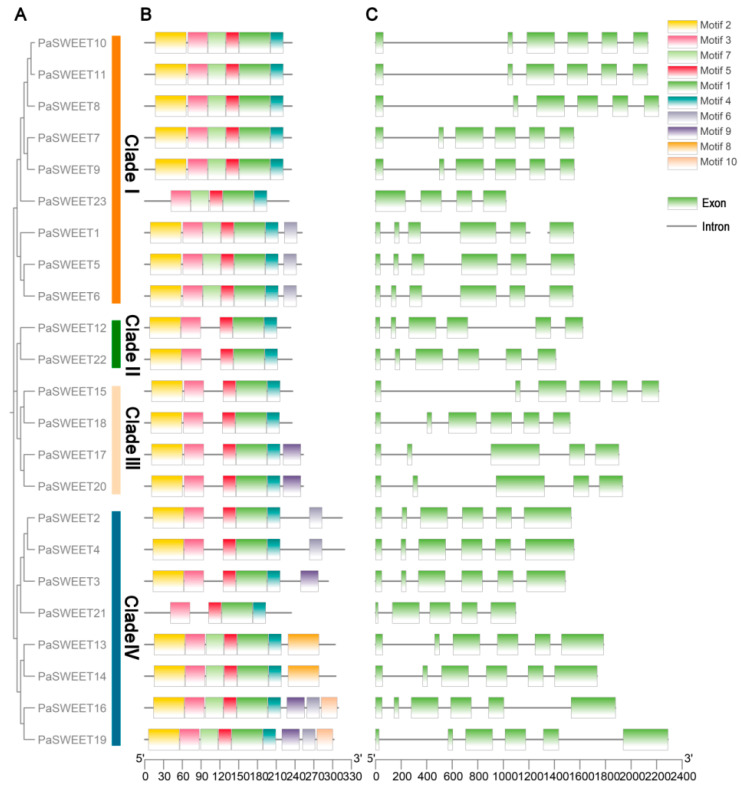
Phylogenetic correlation, conserved motifs, and gene structure analysis of *PaSWEETs*. (**A**) Phylogenetic tree of 23 *PaSWEET* genes based on neighbor joining method and 1000 bootstrap replications. (**B**) Motif compositions of PaSWEET proteins. Ten motifs are specified by different colored boxes. (**C**) Gene structure of *PaSWEET* genes. The associated exons are shown by green boxes, and introns are shown by black lines.

**Figure 4 plants-13-00406-f004:**
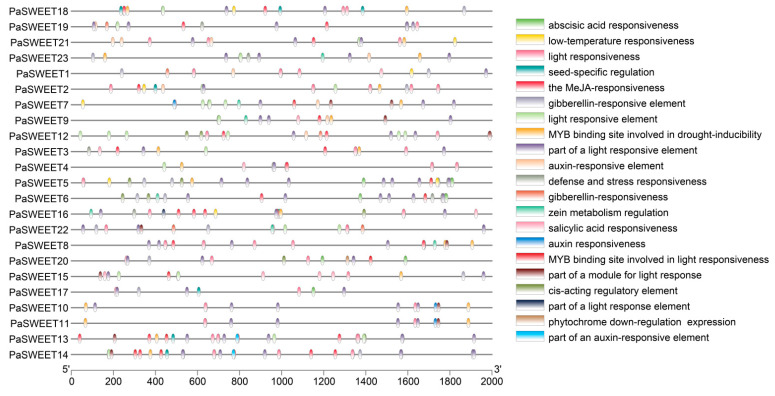
Cis-element analysis in the promoter of *PaSWEETs*. A total of 23 *PaSWEETs* genes having promoter sequences of (200 bp) were analyzed by using the PlantCare database.

**Figure 5 plants-13-00406-f005:**
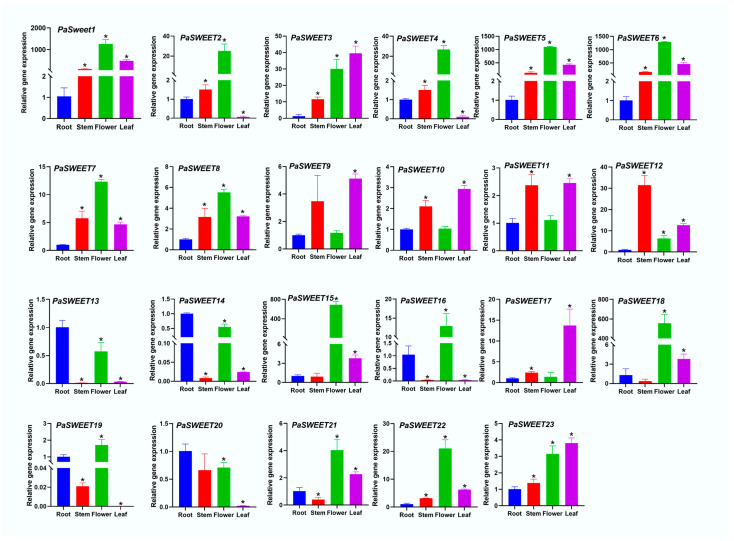
Relative fold gene expression of *PaSWEETs* genes in different sampled tissues. The *X*-axis represents different tissues (roots, stems, flowers, and leaves), and the relative gene expression is shown on *Y*-axis. Asterisks are representing the significant expression results among different tissues.

**Figure 6 plants-13-00406-f006:**
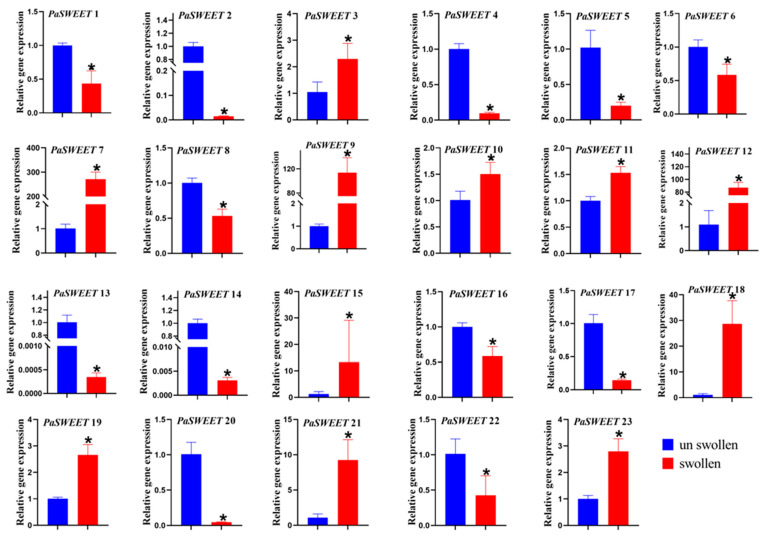
Relative expression levels of *PaSWEETs* genes in swollen and unswollen tubers. The *X*-axis represents swollen (red box) and unswollen tubers (blue box), and the relative gene expression is shown on *Y*-axis. Asterisks are representing the significant expression results between swollen and unswollen tubers.

**Figure 7 plants-13-00406-f007:**
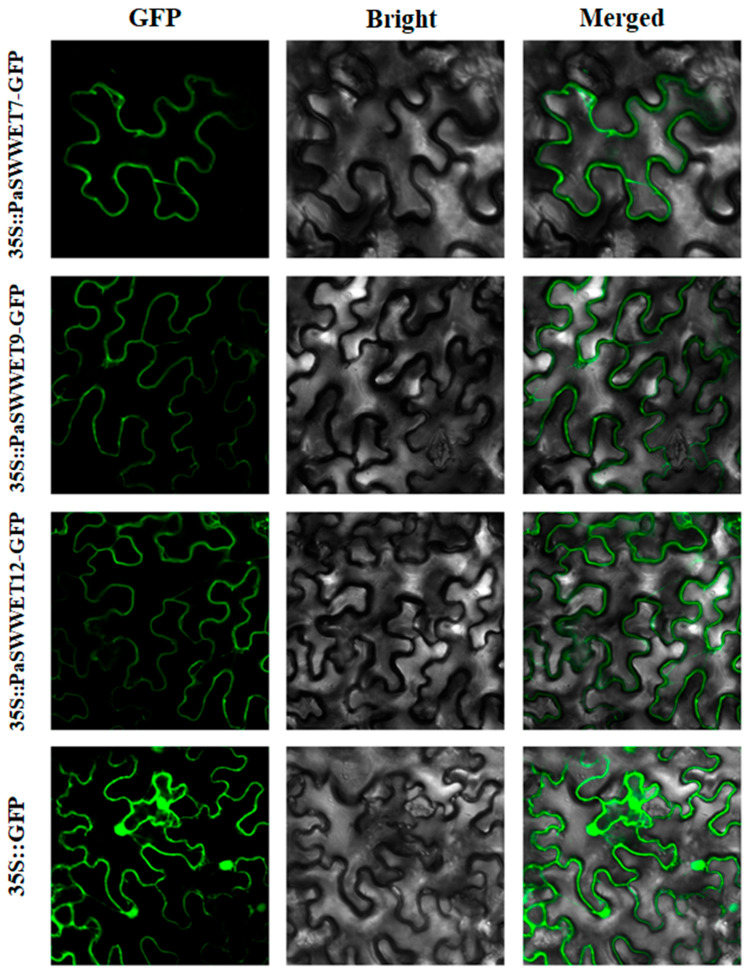
Subcellular localizations of *PaSWEET7*, *PaSWEET9*, and *PaSWEET12* in tobacco leaves. Scale bar = 20 µm. 35S:GFP was used as the control.

**Figure 8 plants-13-00406-f008:**
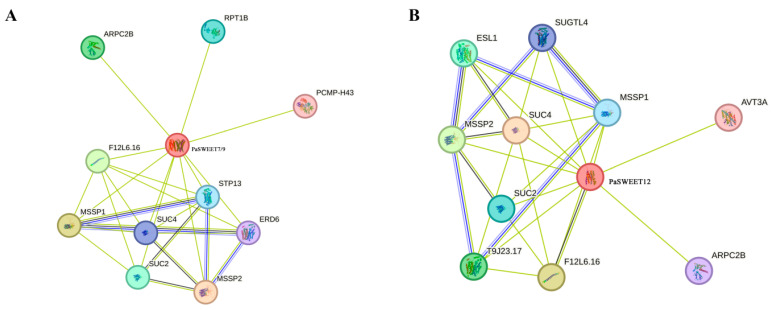
Protein interaction networks of *PaSWEETs* in *P. anserina* and *Arabidopsis thialiana*. (**A**) Protein interaction network of *PaSWEET* (7 and 9). (**B**) Protein interaction network of *PaSWEET12*.

**Table 1 plants-13-00406-t001:** Characteristics of *PaSWEETs* genes in *P. anserina.* Chr (chromosome); CDS (coding sequence); PI (isoelectric point); A.A (amino acids); S.C. (subcellular location); KDa (kilo dalton).

Gene Name	Gene Locus	Start	End	Chr	CDS (bp)	PI	MW (KDa)	A.A	S.C. Location
*PaSWEET1*	Poanv1_3G01215.1	16244178	16245727	Chr3	756	9.62	27.46	251	Cell membrane
*PaSWEET2*	Poanv1_3G02345.1	26445095	26446626	Chr3	948	9.38	35.06	315	Cell membrane
*PaSWEET3*	Poanv1_3G02537.1	27842414	27843899	Chr3	882	5.82	32.7	293	Cell membrane
*PaSWEET4*	Poanv1_4G01502.1	9722164	9723717	Chr4	960	9.23	35.47	319	Cell membrane
*PaSWEET5*	Poanv1_4G02299.1	16809415	16810970	Chr4	753	9.45	27.33	250	Cell membrane
*PaSWEET6*	Poanv1_4G02307.1	16970550	16972095	Chr4	753	9.37	27.32	250	Cell membrane
*PaSWEET7*	Poanv1_5G00681.1	4865439	4866991	Chr5	705	8.95	26.1	234	Cell membrane
*PaSWEET8*	Poanv1_5G01420.1	11911444	11913660	Chr5	708	8.91	26.23	235	Cell membrane
*PaSWEET9*	Poanv1_6G00874.1	6027126	6028681	Chr6	705	8.95	26.18	234	Cell membrane
*PaSWEET10*	Poanv1_6G01829.1	14477140	14479272	Chr6	708	9.08	26.24	235	Cell membrane
*PaSWEET11*	Poanv1_6G01858.1	14918865	14920993	Chr6	708	9.08	26.24	235	Cell membrane
*PaSWEET12*	Poanv1_7G01742.1	22928052	22929674	Chr7	702	8.42	25.94	233	Cell membrane
*PaSWEET13*	Poanv1_7G02056.1	25735871	25737656	Chr7	915	6.52	34.02	304	Cell membrane
*PaSWEET14*	Poanv1_8G01210.1	13751438	13753172	Chr8	918	5.41	33.93	305	Cell membrane
*PaSWEET15*	Poanv1_9G02031.1	23847352	23849568	Chr9	711	9.28	26.46	236	Cell membrane
*PaSWEET16*	Poanv1_9G02352.1	26435793	26437671	Chr9	930	7.03	34.59	309	Cell membrane
*PaSWEET17*	Poanv1_9G02531.1	27818865	27820769	Chr9	762	9.38	28.12	253	Cell membrane
*PaSWEET18*	Poanv1_10G01926.1	21562089	21563611	Chr10	708	9.3	26.49	235	Cell membrane
*PaSWEET19*	Poanv1_10G02196.1	23424599	23426888	Chr10	909	6.93	34.13	302	Cell membrane
*PaSWEET20*	Poanv1_10G02432.1	25180176	25182110	Chr10	762	9.57	28.03	253	Cell membrane
*PaSWEET21*	Poanv1_12G04149.1	41778738	41779835	Chr12	705	9.43	26.68	234	Cell membrane
*PaSWEET22*	Poanv1_13G00740.1	11203434	11204844	Chr13	708	8.59	26.02	235	Cell membrane
*PaSWEET23*	Poanv1_13G02471.1	26252771	26253791	Chr13	693	8.99	26	230	Cell membrane

## Data Availability

The datasets used and analyzed in the current study are available from the corresponding author(s) upon reasonable request.

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
