# Peer review of "Genome-Wide Bioinformatics Analysis of SWEET Gene Family and Expression Verification of Candidate PaSWEET Genes in Potentilla anserina"

_plants, 2024, doi:10.3390/plants13030406_

Round 1
Reviewer 1 Report
Comments and Suggestions for Authors
Very poor quality English. Needs lot of improvement.
Author Response
Response to Reviewer 1 comments;
Title: Genome-wide bioinformatics analysis of SWEET gene family and expression verification of candidate PaSWEET genes in Potentilla anserina.
The manuscript deals with the genome-wide identification of SWEET genes in Potentilla anserina, gene characterization and expressions under various abiotic stress conditions and localization of important genes in the cell organelle of tobacco after transformation. The manuscript has potential and the study is necessary to understand the functions of these SWEET transporters in the swollen and unswollen tuberous roots. However, the manuscript suffers from poor English language throughout. Material and Methods need some improvement.
Answer: First of all, we (the authors) want to say special thanks to the reviewers for their constructive peer review, positive comments and valuable suggestions. We also want to extend our thanks for useful suggestions and comments, which indeed helped to improve the manuscript in a better way. Thanks a lot
Potentilla anserine should be in italics throughout.
Answer: We have done the addressed corrections in the text of the revised manuscript. Thanks
Line 16: Delete the words “activities that” and insert the word “and”.
Answer: We have done the addressed corrections in the text of the revised manuscript. Thanks
Line 54: It should be Musa acuminata (there is a typographical error).
Answer: This correction has been done in the revised manuscript. Thanks
Line 56: Class III (not classes III).
Answer: “Classes III” has been replaced by clade as suggested by other reviewer. Thanks
Line 104: Vitamin C is not an element. Correct the sentence.
Answer: Sorry, We have write Vitamin C by mistake. We have corrected this in the revised manuscript Thanks
Lines 124-125: Delete the brackets in both the lines “PaSWEET4 and PaSWEET12”.
Answer: We have done the addressed corrections in the text of the revised manuscript. Thanks
Line 149: The sub-heading should read as “Phylogenetic analysis of”.
Answer: This correction has been done in the revised manuscript. Thanks
Line 194: The word “In” is repeated. Delete one of them.
Answer: This correction has been done in the revised manuscript. Thanks
Line 225: Rephrase it as “different roles to play in various tissues”.
Answer: We have done the addressed corrections in the text of the revised manuscript. Thanks
Line 235: Replace the word “showed” with “have”.
Answer: This correction has been done in the revised manuscript. Thanks
Lines 238-242: Rephrase the two sentences.
We have done the addressed corrections in the text of the revised manuscript. Thanks
Line 256: Delete the bracket and it should read as “genes 7, 9 and 12”.
Answer: This correction has been done in the revised manuscript. Thanks
Lines 265-268: What is the basis to state that SWEET genes along with other transporters play a vital role in biotic and abiotic stress?
Answer: We have discussed the role of SWEET genes and their associated transporter, in the last highlighted paragraph of discussion section along with citations and references, as per suggestion. Thanks
Line 297: Rephrase it as follows: “PsSWEET gene family members”.
Answer: This correction has been done in the revised manuscript. Thanks
Line 305: Delete the full point (.) before the brackets.
Answer: This correction has been done in the revised manuscript. Thanks
Line 308: It is “abscisic acid” (not Abscisc).
Answer: This correction has been done in the revised manuscript. Thanks
Lines 309-311: Presence of certain hormone-responsive and light-responsive elements in the promoter regions might indicate their role in abiotic stress response, but they do not confirm stress tolerance. So, rephrase the sentence.
Answer: This correction has been done in the revised manuscript. Thanks
Lines 321-322: Rephrase the sentence.
Answer: This correction has been done in the revised manuscript. Thanks
Lines 338: Rephrase as “sweet potatoes, its relative”.
Answer: This correction has been done in the revised manuscript. Thanks
Lines 417-418: Agrobacterium tumefaciens should be in italics.
Answer: This correction has been done in the revised manuscript. Thanks
Lines 418-420: How the tobacco leaves after infection with Agrobacterium were dissected or how the leaf sections were taken is not mentioned. That information may be given in Material and Methods along with magnification details.
Answer: We have done the addressed corrections in the text of the revised manuscript. Thanks
Line 436: Delete the word “verification”.
Answer: This correction has been done in the revised manuscript. Thanks
Reviewer 2 Report
Comments and Suggestions for Authors
Well written paper, based on well-organized, managed and implemented research designs. Two modest issues--one about inferences, another about references. Inferences first--I wondered (and made notes about that wondering) three times in your text--about whether you might be able to suggest the possibility that your research might be relevant to other tuberous crop plants. I encourage you to consider that possibility, particularly given that you do cite a sweet potato reference regarding a similar study (reference #55 or #53, I think). Secondly, please recheck whether your internal reference numbering (within the text) matches the numbering in your reference list. When I went to look for reference #55, I found that it was about radishes rather than about sweet potatoes. Please re-check all of your references.
Finally, note that I have made several comments on the text through the attached document.

Very strong English usage; have made a small handful of suggestions for improvement on the attachment.
Author Response
Replies to Reviewer 2 Comments:
Well written paper, based on well-organized, managed and implemented research designs. Two modest issues--one about inferences, another about references. Inferences first--I wondered (and made notes about that wondering) three times in your text--about whether you might be able to suggest the possibility that your research might be relevant to other tuberous crop plants. I encourage you to consider that possibility, particularly given that you do cite a sweet potato reference regarding a similar study (reference #55 or #53, I think). Secondly, please recheck whether your internal reference numbering (within the text) matches the numbering in your reference list. When I went to look for reference #55, I found that it was about radishes rather than about sweet potatoes. Please re-check all of your references. Finally, note that I have made several comments on the text through the attached document.
Answer: First of all, we (the authors) want to say special thanks to the reviewers for their constructive peer review, positive comments and valuable suggestions. We also want to extend our thanks for useful suggestions and comments, which indeed helped to improve the manuscript in a better way. Thanks a lot
Comment 1: Sound, clear, comprehensive of the main ideas of your paper--good abstract!
Answer: Thanks a lot for your positive comments on our submitted work.
Comment 2: Strong section, firmly establishing the widespread (ubitiquous) nature of these genes.
Answer: Thanks a lot for your positive comments on our submitted work.
Comment 3: Should this word be "class"?
Answer: The word “classes” has been replace by clades, as mentioned by you in other comment. Thanks
Comment 4: Does this imply that SWEETs primarily move sugars via concentration gradients?
Answer: The suggested phenomena have been changed in the revised manuscript and have been highlighted in the second paragraph of introduction section. Thanks
Comment 5: Verb tense/usage awkward here (at least in English). Perhaps "precluding sucrose transport to tubers"?
Answer: We have corrected the text in the revised manuscript, as per suggestion. Thanks
Comment 6: Not tracking with you here--why would sugar leakage control pathogen growth? Would not the free sucrose provide more substrate for the pathogens?
Answer: The above mentioned phenomena have been corrected and highlight in the revised manuscript. Thanks
Comment 7: Really like the way you develop a broad basis for the metabolic and developmental roles of these genes in a wide array of plants.
Answer: Thanks a lot for your positive comments on our submitted work.
Comment 8: Delighted to see that this paragraph focuses on your target organism.
Answer: Thanks a lot for your positive comments on our submitted work.
Comment 9: Italics here?
Answer: The name has been changed to italics. Thanks
Comment 10: Having read to this point, I am wondering if these genes might be important in development of tubers in other species? If you can suggest that possibility, it could potentially enhance the appeal of your manuscript.
Answer: According to your kind suggestion, we have changed it and have been highlighted in the second last paragraph of introduction section. Thanks
Comment 11: Maybe here is where you talk about potential applications of your work to other tuberous species?
Answer: The correction has been done as per your kind suggestion. Thanks
Comment 12: Seems to be a very important finding--right?
Answer: Thanks for appreciation.
Comment 13: This closing point seems to be quite important.
Answer: In our opinion, our study exposed valuable findings that would contribute additional genetic insights for future research. Thank for appreciation.
Comment 14: Another key finding!
Answer: We think that our bioinformatics analysis revealed key finding related to the SWEET gene family and identified potential genes regulating sugar transport in Potentilla anserina. Thanks for kind understanding.
Comment 15: And extra period here, please remove.
Answer: The extra period has been removed. Thanks
Comment 16: I'm hoping that you are able to draw richer inferences about the diverse roles of SWEETs in different tissues in this plant.
Answer: There we only discussed the results, while the diverse roles of SWEETs in different tissues is discussed in the discussion section along with references and has been highlighted. Thanks
Comment 17: Exciting results--I am somewhat confused that the previous paragraph seemed not to touch on the importance of particular SWEETs in tubers; further, that the tuber-active SWEETs do not seem to match the above "root" roster. Am I missing something here?
Answer: (1) The previous paragraph is about the relative gene expression of PaSWEETs in different sampled tissues, and here we disused about relative gene expression of PaSWEETs in swollen and unswollen tubers.
(2) Sorry for the mistake. We upload incorrect figure in the submitted manuscript by mistake. Meanwhile, we have changed the figure (correct) in the revised manuscript. Thanks for understanding
Comment 18: Strong approach to confirm cellular location of these three gene products.
Answer: Thank for appreciation.
Comment 19: I believe that in at least one instance above, you refer to these groups of genes as "classes" rather than as "clades." Should you use "clades" in every instance?
Answer: We have replaced the word “classes” by clades, Thanks
Comment 20: Spelling
Answer: Spelling mistake has been removed. Thanks
Comment 21: Any way that you could be more specific here?
Answer: Here we discussed the relative gene expression in different sampled tissues in our plants, and then it has been supported by previous finding (Highlighted in revised manuscript), along with references. Thanks
Comment 22: italics?
Answer: The name has been changed to italics. Thanks
Comment 23: Does this phrase intend to suggest that your research team did this work with radish?
Answer: we have corrected the text in the revised manuscript, as per suggestion. Thanks
Comment 24: I went looking for this reference in your list--#55 actually appears to be about radishes. Please check correspondence between your text numbering and your reference list numbering. Actually looks like your sweet potato reference might be #53 in the reference list.
Answer: we have corrected text numbering and reference list in the revised manuscript, as per suggestion. Thanks
Comment 25: Glad to see this reference to sweet potatoes (Italics?) here--wondered above if you might be able to suggest extensions of your work with tuberous Pa to other tuberous plants.
Answer: Respected Sir/ Ma’am, Thanks for nice suggestion, we have already added the possible references related to SWEETs gene in tuberous crops, we can’t add more references, Thanks